# Clinical Impact of Rapid Bacterial Microbiological Identification with the MALDI-TOF MS

**DOI:** 10.3390/antibiotics12121660

**Published:** 2023-11-25

**Authors:** Miriam Uzuriaga, José Leiva, Francisco Guillén-Grima, Marta Rua, José R. Yuste

**Affiliations:** 1Clinical Microbiology Service, Clínica Universidad de Navarra, 31008 Pamplona, Spain; miriamuzuriaga@hotmail.com (M.U.); mrua@unav.es (M.R.); 2Healthcare Research Institute of Navarre (IdiSNA), 31008 Pamplona, Spain; frguillen@unav.es (F.G.-G.); jryuste@unav.es (J.R.Y.); 3Department of Preventive Medicine, Clínica Universidad de Navarra, 31008 Pamplona, Spain; 4CIBER in Epidemiology and Public Health (CIBERESP), Institute of Health Carlos III, 46980 Madrid, Spain; 5Department of Health Sciences, Public University of Navarra, 31008 Pamplona, Spain; 6Service of Infectious Diseases, Clínica Universidad de Navarra, 31008 Pamplona, Spain; 7Department of Internal Medicine, Clínica Universidad de Navarra, 31008 Pamplona, Spain

**Keywords:** clinical impact, quick information, diagnostics, MALDI-TOF MS, antibiotic use

## Abstract

Rapid microbiological reports to clinicians are related to improved clinical outcomes. We conducted a 3-year quasi-experimental design, specifically a pretest–posttest single group design in a university medical center, to evaluate the clinical impact of rapid microbiological identification information using MALDI-TOF MS on optimizing antibiotic prescription. A total of 363 consecutive hospitalized patients with bacterial infections were evaluated comparing a historical control group (CG) (n = 183), in which the microbiological information (bacterial identification and antibiotic susceptibility) was reported jointly to the clinician between 18:00 h and 22:00 h of the same day and a prospective intervention group (IG) (n = 180); the bacterial identification information was informed to the clinician as soon as it was available between 12:00 h and 14:00 h and the antibiotic susceptibility between 18:00 h and 22:00 h). We observed, in favor of IG, a statistically significant decrease in the information time (11.44 h CG vs. 4.48 h IG (*p* < 0.01)) from the detection of bacterial growth in the culture medium to the communication of identification. Consequently, the therapeutic optimization was improved by introducing new antibiotics in the 10–24 h time window (*p* = 0.05) and conversion to oral route (*p* = 0.01). Additionally, we observed a non-statistically significant decrease in inpatient mortality (global, *p* = 0.15; infection-related, *p* = 0.21) without impact on hospital length of stay. In conclusion, the rapid communication of microbiological identification to clinicians reduced reporting time and was associated with early optimization of antibiotic prescribing without worsening clinical outcomes.

## 1. Introduction

The inappropriate use of antibiotics is a major concern within healthcare systems, and it contributes to antimicrobial resistance, increased healthcare costs, and reduced quality of care [1,2]. Previous studies have revealed that over 50% of antibiotic prescriptions are unnecessary or inappropriate [3], and this leads to antimicrobial resistance (AMR) [4]. Moreover, the World Health Organization (WHO) predicts it may increase in the coming years, becoming one of the most significant public health hazards [4]. For these reasons, it highlights the urgent need for improved strategies to optimize antibiotic use [3,5,6].

Several studies have demonstrated that early and accurate microbiological diagnosis can significantly improve antibiotic prescribing practices and reduce inappropriate antibiotic utilization [7,8].

Traditionally, bacterial identifications in clinical microbiology laboratories are based on phenotypic characteristics, mainly biochemical reactions, employing manual or automated systems. One of the classically used and most widespread automated systems is the Vitek 2 (bioMerieux) system. In the last decade, the matrix-assisted laser desorption/ionization time-of-flight mass spectrometry (MALDI-TOF MS) technique has gained popularity in microbiology laboratories, which, by analyzing protein fingerprints from whole bacterial cells, allows for the identification of bacteria and yeasts in a faster and easier way [9,10,11,12,13]. The MALDI-TOF MS system has revolutionized clinical microbiology laboratories by reducing the time required for microbial identification, enabling faster diagnoses, and optimizing antibiotic usage [14,15,16]. The use of MALDI-TOF MS is widely known, used not only on bacteria but also on fungi [17,18] and mycobacteria [19]. In addition, it is being used to understand bacterial resistance [20]. Additionally, the incorporation of MALDI-TOF MS in clinical laboratories has the potential to reduce mortality rates in blood cultures and shorten hospital stays in intensive care units [21,22,23,24].

Several authors have reported the presence of a positive impact of early information from MALDI-TOF MS data, but these are limited to pathologies and bacteria isolated from specific samples, most frequently with blood [10,25], urine [10,26], cerebrospinal fluid [27], and ascitic fluid [28]. However, there are no studies with such variability of samples. Based on these findings, this study aims to evaluate the clinical impact of early microbiological identification information using MALDI-TOF MS on optimizing antibiotherapy, including the total number of samples processed for bacteriological culture in the microbiology laboratory.

## 2. Results

We enrolled 363 consecutive hospitalized patients with bacterial infections and divided them into two groups: a control group (CG) with 183 patients and an intervention group (IG) with 180 patients.

### 2.1. Demographics and Clinical Basal Characteristics of Patients

The main features of the patients are shown in Table 1. In both groups, McCabe–Jackson criteria (severity of underlying illness) and the Charlson score (comorbidity index) were similar (CHI). The severity of infections and the place of acquisition of infection were also identical. According to the department in charge, medical departments are the most frequent in both groups, but surgical departments are more frequent in IG (22 patients in CG vs. 37 patients in IG (*p* = 0.01)).

### 2.2. Samples and Infectious Syndrome

The average time from sample collection to report by telephone and by computerized laboratory information system of the identification of index-positive culture (IPC) to the clinician was significantly shorter (8.32 h) in the IG compared to the CG (29.93 h vs. 38.25 h *p* < 0.01). IPC is defined as the first sample with bacterial growth. Additionally, the mean time from the detection of bacterial growth to the communication of identification results to the clinician was significantly shorter (6.96 h) in the IG compared to the CG (4.48 h vs. 11.44 h, *p* < 0.01). However, no significant differences were observed in the time elapsed between processing the sample and the detection of bacterial growth between the groups (24.96 h in IG vs. 25.63 h in CG, *p* = 0.14).

Regarding infectious syndromes, the number of patients with a clinical diagnosis of pneumonia was significantly higher in the CG (*p* < 0.01), and skin and soft tissue infections were more frequent in the IG (*p* = 0.03) (Table 1).

The most frequently processed samples corresponded to urine, lower respiratory tract, blood culture, and wounds and abscesses. There were no significant differences between the groups related to the type of IPC (Table 1). However, associated with the isolated microorganisms, *Staphylococcus aureus* was more prevalent in the IG (26 (11.8%) isolates) than in CG (10 (4.5%) isolates); *p* = 0.01. (Table 2). The total number of bacteria isolated in the CG was 223, while in the IG, there were 219 from IPC. In the CG, 41 (22.4%) attendances were polymicrobial, while in the IG, there were 38 (21.1%) attendances (*p* = 0.76).

### 2.3. Outcomes in Antibiotic Therapy of Early Microbiological Information

Conversion from parenteral to the oral route was significantly higher in IG (15 (13.9%) vs. CG (10 (8.1%); *p* = 0.01) as well as from parenteral to no treatment (CG 2 (1.0%) vs. IG 4 (2.2%); *p* = 0.03).

In the 0–10 h and 10–24 h time slots, a higher number of cases of addition of one or more antibiotics compared to previous antibiotic therapy was observed in CG (*p* = 0.01 and 0.09, respectively); in the 10–24 h time window, a higher number of cases of the instauration of antibiotics was observed in the IG compared to the CG (*p* = 0.05), and in the 24–36 h time slot, the elimination of at least one antibiotic was higher in the CG (*p* = 0.05). Finally, beyond 36 h, both groups had no differences in antibiotic optimization. All other therapeutic modifications (antimicrobial substitution, elimination, addition, or initiation of antibiotic treatment) in antimicrobial therapy were similar in both groups (Table 3).

The most frequent type of therapeutic modifications in both groups was “substitution” (74 (46.2%) of 160 total modifications in CG and 77 (52.0%) of 148 in IG), followed by “instauration” of antibiotic treatment for the first time, “elimination” of at least one antibiotic, and “addition” of at least one antibiotic (Table 3). The percentage of patients who received adequate empirical therapy based on antimicrobial susceptibility was 87.6% (86.9% in CG vs. 88.3% in IG, *p* = 0.75).

When Gram-positive bacteria were isolated in IPC, the number of changes in the early window time of 10–24 h was significantly higher in IG (*p* ≤ 0.01) with an instauration of antibiotics close to significance (*p* = 0.07).

### 2.4. Hospital Impact and Mortality Outcomes

Hospital times were evaluated from IPC until the patient was discharged or died during hospital admission. There was no significant difference in the median length of hospitalization for the IG compared to the CG (10.72 days in the CG vs. 11.14 days in the IG, *p* = 0.91). Additionally, no differences were observed in the median duration of hospitalization in the special hospitalization area (intermediate care beds) (3.95 days in CG vs. 6.21 days in IG, *p* = 0.47), in the ICU (2.81 days in CG vs. 5.96 days in IG, *p* = 0.29), and in the conventional hospital stay (9.75 days in CG vs. 9.87 days in IG, *p* = 0.97) (Table 4).

No significant differences were observed in the duration of endotracheal intubation between the CG (3 days, 12 patients) and the IG (4.5 days, 12 patients) (*p* = 0.62), nor the duration of non-invasive mechanical ventilation between the CG (27 days, 15 patients) and the IG (4 days, 15 patients) (*p* = 0.38). The mortality rate during admission did not show significant differences between both groups: 31 (17.2%) patients in the CG vs. 21 (11.5%) patients in the IG (*p* = 0.15). However, the IG exhibited a reduction in mortality related to infections, but the difference in infection-related mortality between the CG (14 (7.8%) and the IG (8 (4.3%)) was not statistically significant. Mortality at 30 days was 0% in CG and 1.6% in patients in IG, of which two were infection-related (1.1%), and the CHI was above the median (CHI 7 and 9) (Table 4).

In the subanalysis with the Gram-positive bacteria, a reduction in hospital times was found, decreasing the total stay time (*p* = 0.04) and, consecutively, the conventional stay time (*p* ≤ 0.01) in the IG. In turn, the ICU admission time decreases once the sample has been identified (*p* = 0.03).

In another subanalysis, in patients with positive blood cultures (IG 39 patients (57.3%) compared to CG 29 patients (42.7%)), decreasing overall 30-day mortality with six (21%) deaths in CG compared to two (5%) deaths in IG; *p* = 0.04.

## 3. Discussion

This study evaluates the clinical impact of early microbiological diagnosis in a real-world scenario. Our findings suggest that the reduced turnaround time of microbiological procedures and the possibility of early identification allow for the early optimization of antimicrobial treatment. Most published studies are based on limited samples such as blood [25] or urine cultures [26]. Our study evaluates the clinical impact of early microbiological diagnosis in different clinical samples, as shown in Table 2.

A previous study was conducted by our group [3,5], in which we compared the identification and antimicrobial susceptibility information with Vitek 2 (bioMerieux) at 18:00 h–22:00 h with the same information provided the following day. A reduction in identification and antimicrobial susceptibility test time of 17.4 h was reported, which had a significant impact in terms of reduced therapeutic intervention time, hospital stay, ICU stay, and assisted ventilation requirement. In contrast, our current study compared the identification information with Vitek 2 (bioMerieux) in CG and Vitek MS (bioMerieux) in IG with a reduction time of 8.32 h. The identification time of MALDI-TOF MS is shorter than that of Vitek 2 (bioMerieux), reducing the time required to report identification results. The MALDI-TOF MS system also allows greater precision and speed in microbial identification thanks to continuously updating the existing bacterial protein spectra [29]. As in the previous study, both groups maintained the antimicrobial susceptibility information between 18:00 h and 22:00 h.

The speed in obtaining and transmitting the information is due to the fast identification process. This rate of obtaining and transmitting results to clinicians could benefit patients suffering from severe infectious diseases requiring immediate therapeutic action. Moreover, by working with such a wide range of samples, we can realistically study the clinical impact of rapid bacterial identification information.

This reduction time response resulted in an increased conversion from intravenous to oral route antibiotics and a non-significant decrease in overall mortality in the IG, indicating a smaller magnitude of difference.

Some authors, such as Bellido et al. [13], emphasize the need to consider other more sensitive and early alternatives, such as Gram staining as rapid microbiological information for early optimization of antibiotic treatment. In contrast, our study bases rapid information of microbiological identification using a precise technique such as MALDI-TOF MS. The information on the Gram stain is available before the microbiological identification and could help explain the lower impact obtained in our study compared to those based on the Gram stain [5]. However, authors such as Torres et al. [30] consider that despite the early availability of Gram stain, MALDI-TOF MS is a valuable tool that can minimize errors in empirical antibiotic therapy by fine-tuning antibiotics according to the diagnosis.

Gram staining allows for an earlier diagnostic approach, as it allows for optimization of treatment intervention prior to MALDI-TOF MS results, which may have a reduced impact on our study. In our study, this interval ranged from 4 to 8 h. This temporary reduction may also explain a lower impact on our results because if the patient does not present an unfavorable evolution, some clinicians may decide to wait for susceptibility information before making changes to treatment.

We have successfully demonstrated the clinical optimization that MALDI-TOF MS can provide daily. Our findings are consistent with other studies [1,13,14,31], which describe the impact of MALDI-TOF MS on reducing diagnosis time, optimizing antibiotic treatment, and potentially decreasing mortality, although not significantly.

Our study significantly impacted highly virulent organisms such as *S. aureus*, previously described by Samaranayake et al. [14]. Although we did not find statistically significant differences in the overall reduction in hospital stay, we observed significant differences in total stay times and ICU hospitalization time in patients with positive blood cultures, where Gram-positive microorganisms tend to be predominant. Recent studies indicate that adjusted antibiotic treatment rather than empirical broad-spectrum antibiotic therapy prevents the development of resistance [32]. Many clinicians tend to continue with empirically prescribed antibiotic therapy even after receiving microbiological reports to maintain the efficacy of the chosen antibiotic regimen [22,23,24]. Our study’s empirical treatment was effective based on the antibiotic susceptibility test results (86.9% in CG vs. 88.3% in IG), identifying pathogens that affected antibiotic therapy modification, with 84.69% in CG and 82.22% in IG switching from the previous antibiotic regimen.

On the other hand, there is an increasing rate of antibiotic resistance in our environment, where markers for the study period were MRSA resistance 15%, *Escherichia coli* ESBL-producing 9%, *E. coli* AmpC plasmid 4%, *E. coli* carbapenems resistance 0%, *Klebsiella pneumoniae* ESBL-producing 13%, *K. pneumoniae* carbapenems resistance 3*%, K. pneumoniae* AmpC plasmid 7%, *Pseudomonas aeruginosa* imipenem resistance 20%, and *P. aeruginosa* meropenem resistance 12%. This resistance scenario may motivate the clinician, once adequate empirical therapy has been established in both groups (87%), to wait to know the results of antimicrobial sensitivity to introduce any therapeutic change, especially when the sensitivity to antimicrobial agents will be available on the same day.

In our study, the prompt provision of information and identification via telephone and computer, using a laboratory information system, enabled clinicians to adjust antibiotic treatment and administer effective oral or combined oral and intravenous antibiotherapy (*p* = 0.01), reducing exclusive intravenous administration (*p* = 0.03). This finding aligns with the study conducted by Mertz et al. [33], highlighting the ability of early diagnosis to optimize therapeutic approaches. Additionally, it is worth noting that the rapid microbiological information led to significant changes within the time slot close to identification (10–24 h) with *p* = 0.05, involving the introduction of new antibiotics in the IG compared to the CG. These findings are consistent with those reported by Perez-Lopez [34], emphasizing that rapid reporting contributes to the optimization of antibiotherapy. Our results align with studies conducted by de la Pedrosa [8], Osthoff [[35], and Porreca [36], which describe the clinical impact of reducing the time for identification information and employ a similar methodology to our study. Notably, we report the antibiogram at 6–8 h after antimicrobial identification, while Osthoff [35], which takes the shortest of the three studies cited, takes 10 h. Other studies have demonstrated a statistically significant reduction in overall mortality and mortality related to infections with rapid diagnosis [37,38]. These studies exhibited a notable decrease in the percentage of patients who died overall and deaths attributed explicitly to infections. Our study observed lower overall mortality and infection-related mortality in the IG, although the differences were not statistically significant. This could be attributed to Gram stain information being reported promptly in both groups. The early availability of this information may have diminished the impact of infection-related mortality in both groups, as appropriate antibiotic prescriptions based on microscopic findings were initiated early to expedite treatment and reduce mortality. Gram staining provides valuable and rapid information, but it may be insufficient for selecting the initial antibiotic treatment in cases involving the selection of antibiotic treatment of the different samples, especially in blood cultures [30].

These results are consistent with those obtained in previously published studies by Roux [7], Huang [21], and Niwa [22], which described how early diagnosis of bacterial identification improved time to effective antibiotic therapy and adjustment of antimicrobial treatment. Additionally, Osthoff [35] reported that early diagnosis contributed to lower overall mortality rates in the IG 17 compared to CG (*p* = 0.6). Regarding 30-day mortality (9.6 vs. 16.4%, *p =* 0.06), they were numerically lower in the MALDI-TOF MS group. No reference was made to infection-related mortality. Mortality, although decreasing, does not decrease significantly, probably because most patients are not critically ill. The number of patients requiring ICU was not high, so looking at a larger sample of critically ill patients would be necessary to find statistically significant differences.

Mortality in patients with *S. aureus* bacteremia also occurs in other studies, such as D. Bai’s systematic review [39], obtaining a mortality rate of 27% per month, as well as that of Lewis Pharms [40], which found 18%, while we found a mortality rate of 21% in CG; however, we found a mortality rate of 4% in IG.

Furthermore, the absence of findings regarding infection-related mortality could be due to the limited sample size available to represent such data. With a larger sample, it might be possible to identify these differences. Our study has some limitations, such as its observational design, the fact that its data were analyzed retrospectively, and the comparison between a prospective intervention group and a historical control group since the lack of random assignment can introduce confounding variables that might influence the results, such as the introduction of new antibiotics and different clinical practices in the context of medical updates in the IG on GC. However, the availability of updated information enabled a comparison between the two groups, revealing similarities in clinical outcomes, such as hospital stay and both mortality rates global and related to infection. With a larger sample size, it might be possible to identify significant differences in infection-attributable mortality.

## 4. Materials and Methods

### 4.1. Setting

This study was conducted in the Clínica Universidad de Navarra (Pamplona, Spain), a 300-bed university medical center. The Service of Clinical Microbiology conducted it at Clínica Universidad de Navarra (Pamplona, Spain) in collaboration with the Service of Infectious Diseases. An Infectious Disease clinician evaluated all patients.

All eligible patients were consecutively enrolled, and the number of patients with a positive urine culture was limited to approximately one-third of the samples in both groups (31.7%).

We included 363 hospitalized patients with documented bacterial infections and confirmed bacterial isolations into two groups: a CG and an IG. The CG, whose collection period was from June 2014 to December 2015, included 183 patients in whom the microbiological information of bacterial identification and antibiotic susceptibility was reported simultaneously to clinicians between 18:00 h and 22:00 h of the same day of the detection of bacterial growth (IPC). The IG, whose collection period was from January 2016 to September 2017, included 180 patients in whom the microbiological information of bacterial identification was reported to the clinician as soon as it became available between 12:00 h and 14:00 h (rapid information), while the information on the antibiotic susceptibility data was provided when it was available, between 18:00 h and 22:00 h (Figure 1).

This study uses a quasi-experimental design, specifically a pretest–posttest single-group design. The study is longitudinal and prospective. The CG represents the “pretest” period (the standard procedure before the intervention was applied), and the IG represents the “posttest”period (after the intervention was applied).

The identification of the CG data was compared using the Vitek 2 (bioMerieux) system, which allows the identification of the positive index culture (PIC) and the antibiogram independently using the same device. In contrast, in the IG, the identification was carried out using Vitek MS (bioMerieux), reducing identification time. Antibiotic susceptibility information was performed between 18:00 h and 22:00 h in both groups using Vitek 2 (bioMerieux, Marcy l’Etoile, France).

In the CG, samples were cultured on agar plates. We transferred the colony into a bacterial suspension with the inoculating loop to a defined McFarland of 0.5. This dilution was carried out for identification by the Vitek 2 (bioMerieux) system using the identification cards used for Gram-positive bacteria, GP, and for Gram-negative bacteria, GN. For the antimicrobial susceptibility testing, AST-243 and AST-244 cards were used for *Enterobacteriaceae*, AST-245 for *Pseudomonas* and nonfermenting Gram-negative bacilli, AST-589 for *Enterococcus*, and AST-626 for *Staphylococcus*. The card was automatically filled by a vacuum device, sealed, and inserted into the Vitek 2 (bioMerieux) reader-incubator module (incubation temperature 35.5 °C). The results were analyzed using a database and were extracted automatically.

In IG, identification was performed by Vitek MS (bioMerieux), an analysis of the protein spectrum generated by the bacteria (IPC), and its spectrum was compared with the existing protein profiles in the database, reducing identification times. The samples were inoculated on the following culture media: blood agar, MacConkey, chocolate, CPS in aerobiosis, and blood agar in anaerobiosis as appropriate. The colonies developed on the culture media were deposited onto the sample spots on the target slide model Flexi-Mass-DS TO-430 (bioMérieux). Using a micropipette, we applied 1 μL of VITEK MS-CHCA matrix (bioMérieux) to each smear on the sample and air-dried the mixture until the matrix and sample cocrystallized. Onto the sample spots central on the target slide, *E. coli* strain ATCC 8739 was applied as a calibrator of the kit. Then, the slide with all the prepared samples was loaded into the VITEK MS system to acquire the mass spectra of all the bacterial cell proteins, composed mainly of ribosomal protein, for each sample. The results obtained were analyzed using SARAMIS-KB-V4-17-0 software and expressed by generating spectra.

The Clínica Universidad de Navarra ethics committee approved the study with the number 154/2014 on 8 January 2015.

### 4.2. Power Analysis and Sample Size Calculation

Sample size calculations were performed using Siz version 2.0 software from Cytel Software, based on data obtained from Galar et al. [3]. In their study, the mean cost per patient following early detection using current procedures was EUR 12,402, with a standard deviation of EUR 11,087. A total of 338 patients, comprising 169 patients per group, would be required to detect a decrease in average expenditure per patient of EUR 3000 (24%), with a 95% confidence level and 80% statistical power. This power analysis aimed to ensure that our study was adequately powered to detect statistically significant differences between groups, thereby minimizing Type II errors.

### 4.3. Statistical Analysis

The data were analyzed using IBM SPSS Statistics version 20. Descriptive statistics were calculated for all the variables. For continuous variables, this included the mean, standard deviation (SD), maximum, minimum, and interquartile range (first quartile—Q1 and third quartile—Q3). For categorical variables, proportions were reported to provide a comprehensive understanding of the distribution of each category.

The normality of data was assessed using three different tests: skewness, kurtosis, and the Shapiro–Wilk test. Skewness and kurtosis statistics were used to evaluate the distribution’s symmetry and tailedness, respectively. A Shapiro–Wilk test with a non-significant result (*p* > 0.05) suggested a normal distribution. All assumptions of each test were verified before proceeding with the respective analyses. In cases of normal data distribution, Students’ *t*-tests were employed to compare means between two groups. For data that did not follow a normal distribution, non-parametric Mann–Whitney U tests were used. These tests were chosen to match the nature of our data and ensure the most reliable results.

Chi-square tests were conducted for categorical variables to determine any significant association between them. The level of statistical significance for all tests was set at *p* < 0.05. All *p*-values are two-tailed, reflecting the non-directional nature of our tests and suggesting statistical significance for differences in both directions [41].

### 4.4. Variables Recorded Included

The demographics and clinical variables of the patients evaluated were age, gender, and the severity of the underlying disease according to the McCabe–Jackson criteria [42] and the Charlson comorbidity score [43]. Other variables comprised the severity of the infection, the department in charge, the place of the acquisition of the infection, the index-positive culture, and the infectious syndrome.

To assess the clinical impact of rapid reporting of microbiological identification, we examined the total stay from patient admission and the report of the positive index culture. The entire stay was analyzed according to a hospital stay in a conventional inpatient unit (non-intensive care unit), a special hospitalization area (critically ill patients without the need for vasoactive medications and not requiring orotracheal intubation), and an intensive care unit. Duration of invasive and non-invasive mechanical ventilation and mortality (global and infection-related mortality rates) were also evaluated.

Response times were also analyzed as time from culture positive index to telephone reporting was analyzed. The study also recorded the time of sample collection, arrival, and processing of samples and the timing of oral and automated reports of antimicrobial identification and susceptibility test results.

To evaluate the impact of early information in the optimization of antibiotic therapy, the antibiotic administration at specific time points (0, 10, 24, 24, 36, 48, and 96 h relative to the IPC) was examined in terms of antibiotic changes, including substitution, elimination at intake, addition and no changes.

## 5. Conclusions

Early communication of microbiological identification provided to clinicians was associated with reduced reporting time and early optimization of antibiotic prescribing (increased oral sequencing and change to antibiotic with spectrum adjusted) without compromising clinical factors such as mortality and length of stay.

The low impact could be justified because the time elapsed between information on identification and antimicrobial susceptibility test was relatively short, and 87.5% of the patients had adequate antibiotic treatment when reporting the bacterial identification information.

## Figures and Tables

**Figure 1 antibiotics-12-01660-f001:**
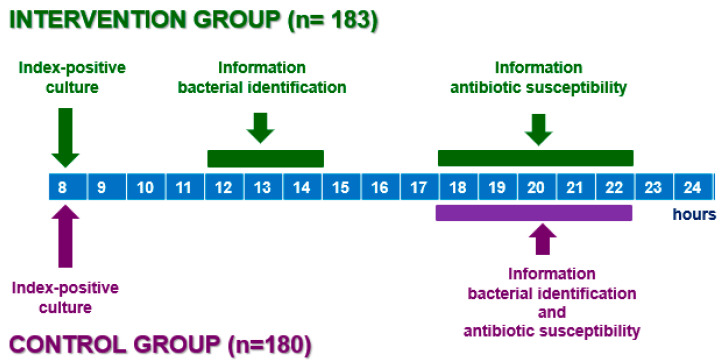
Microbiological information algorithm, performance times in the control group and intervention group.

**Table 1 antibiotics-12-01660-t001:** Demographic, clinical, and microbiological basal characteristics of patients.

	Control Group n = 183	Intervention Group n = 180	Total n = 363	*p*-Value
Mean age ± SD, years	65.9 ± 16.5	65.5 ± 18.1	65.7 ± 17.3	0.85
(range)	(5–96)	(0–97)	(0–97)	
Male Gender, no. (%)	105 (57.4)	118 (65.6)	223 (61.4)	0.11
McCabe–Jackson, no. (%)				
Non-fatal	53 (29)	69 (38)	122 (34)	0.06
Rapidly fatal	25 (14)	22 (12)	47 (13)	0.68
Ultimately fatal	105 (57)	89 (49)	194 (53)	0.13
Charlson index, median (Q1–Q3)	4.2 (2–6)	3.8 (1.25–6)	4.0 (2–6)	0.54
Severity of Infection, no. (%)				
No Sepsis	162 (88.5)	167 (92.8)	329 (90.6)	0.16
Sepsis	12 (6.5)	9 (5)	21 (5.8)	0.52
Septic Shock	9 (5)	4 (2.2)	13 (3.6)	0.16
Department in charge, no. (%)				
Medical	139 (76.0)	121 (67.2)	260 (71.7)	0.06
Surgical	22 (12.0)	37 (20.6)	59 (16.2)	0.01
Medical-surgical	22 (12.0)	22 (12.2)	44 (12.1)	0.95
Place of acquisition, no. (%)				
Community	57 (31.1)	60 (33.3)	117 (32.2)	0.69
Nosocomial	75 (41)	73 (40.6)	148 (40.8)	0.92
Healthcare related	51 (27.9)	47 (26.1)	98 (27)	0.69
Index positive culture, no. (%)				
Biological fluids *	12 (6.6)	9 (5)	21 (5.8)	0.55
Biopsies	6 (3.3)	5 (2.8)	11 (3)	0.76
Blood cultures	28 (15.3)	39 (21.7)	67 (18.5)	0.11
Urine	59 (32.2)	56 (31.1)	115 (31.7)	0.84
Respiratory	48 (26.2)	32 (17.7)	80 (22)	0.06
Prosthesis	2 (1.1)	0 (0)	2 (0.5)	0.16
Wounds and abscesses	28 (15.3)	39 (21.7)	67 (18.5)	0.11
Infectious syndrome, no. (%)				
Endovascular infections				
Catheter site infection	2 (1.1)	1 (0.6)	3 (0.8)	0.57
Endocarditis vascular	2(1.1)	0 (0)	2 (0.6)	0.16
Abdominal				
Intraabdominal	18 (9.8)	13 (7.2)	31 (8.5)	0.37
Biliary	6 (3.3)	4 (2.2)	10 (2.8)	0.55
Respiratory				
Pneumonia	23 (12.6)	8 (4.4)	31 (8.5)	0.01
Upper respiratory tract	0 (0)	2 (1.1)	2 (0.6)	0.15
Lower respiratory tract (not pneumonia)	29 (15.8)	38 (21.1)	67 (18.4)	0.20
Bone and joint	2 (1.1)	7 (3.9)	9 (2.5)	0.09
Skin and soft tissue	25 (13.7)	40 (22.2)	65 (17.9)	0.03
Central nervous system	1 (0.5)	1 (0.6)	2 (0.6)	0.76
Urinary	74 (40.4)	63 (35.0)	137 (37.7)	0.28
No focus	1 (0.6)	3 (1.7)	4 (1.1)	0.31

* Biological fluids: ascitic fluid, pleural fluid, articular fluid, biliary fluid.

**Table 2 antibiotics-12-01660-t002:** Microorganisms isolated from index-positive culture.

	Bacterial Isolate	Control Group n = 183	Intervention Group n = 180	*p*-Value
**Gram-positive**	*Staphylococcus aureus*	10 (4.5)	26 (11.9)	0.01
	CNS	16 (7.2)	7 (3.2)	0.06
	*Enterococcus* spp.	40 (17.9)	36 (16.4)	0.67
	*Streptococcus* spp.	5 (2.2)	6 (2.7)	0.73
	Other Gram-positive	2 (0.9)	3 (1.4)	0.64
**Anaerobes**		4 (1.8)	6 (2.7)	0.50
**Gram-negative**	*Enterobacteriaceae*	104 (46.6)	97 (44.3)	0.62
	Non fermenter	34 (15.2)	34 (15.5)	0.94
	Other Gram-negative	8 (3.6)	4 (1.8)	0.25
**Total isolates**		**223**	**220**	

CNS: Coagulase-negative staphylococci.

**Table 3 antibiotics-12-01660-t003:** Antibiotic changes according to time from index-positive culture.

	0–10 h			10–24 h			24–36 h			36–48 h			48–96 h		
No. (%)	CG	IG	*p*	CG	IG	*p*	CG	IG	*p*	CG	IG	*p*	CG	IG	*p*
Substitution	31	29	0.83	21	22	0.82	5	6	0.74	12	9	0.52	5	11	0.11
	(16.9)	(16.1)		(11.5)	(12.2)		(2.7)	(3.3)		(6.6)	(5.0)		(2.8)	(6.1)	
Instauration	20	25	0.39	1	6	0.05	1	1	0.99	2	0	0.16	1	2	0.55
	(10.9)	(13.9)		(0.5)	(3.3)		(0.5)	(0.6)		(1.1)			(0.5)	(1.1)	
Elimination	12	14	0.65	5	6	0.74	4	0	0.05	2	1	0.57	4	5	0.42
	(6.6)	(7.8)		(2.7)	(3.3)		(2.2)			(1.1)	(0.6)		(2.2)	(2.8)	
Addition	18	6	0.01	7	2	0.09	1	0	0.32	5	2	0.26	3	1	0.32
	(9.8)	(3.3)		(3.8)	(1.1)		(0.5)			(2.7)	(1.1)		(1.6)	(0.6)	
Total changes	81	74	0.54	34	36	0.73	11	7	0.35	21	12	0.11	13	19	0.24
	(44.3)	(41.1)		(18.6)	(20.0)		(6.0)	(3.9)		(11.5)	(6.7)		(7.1)	(10.6)	
No changes	102	106	0.54	149	144	0.73	172	173	0.35	162	168	0.11	170	161	0.24
	(55.7)	(58.9)		(81.4)	(80.0)		(94.0)	(96.1)		(88.5)	(93.3)		(92.9)	(89.4)	
Conversion to oral route	3	7	0.19	4	2	0.42	1	1	0.99	1	2	0.55	1	3	0.31
	(1.6)	(3.8)		(2.2)	(1.1)		(0.5)	(0.5)		(0.5)	(1.1)		(0.5)	(3.8)	

CG: control group; IG: intervention group.

**Table 4 antibiotics-12-01660-t004:** Clinical outcomes.

	Control Group	Intervention Group	*p*-Value
Patients in total say no. (%)	183 (100)	180 (100)	1
Length of the total, say, median ± SD (days)	10.72 ± 16.69	11.14 ± 48.5	0.91
Patients in CHU, no. (%)	178 (97.3)	172 (95.5)	0.39
Length of stay in CHU, median ± SD (days)	9.75 ± 16.33	9.87 ± 48.24	0.97
Patients in special hospitalization area, no. (%)	16 (8.7)	18 (10)	0.68
Length of stay in special hospitalization area, median ± SD (days)	3.95 ± 5.49	6.21 ± 11.42	0.47
Patients in ICU, no. (%)	40 (21.8)	26 (14.4)	0.06
Length of stay in ICU, median ± SD (days)	2.81 ± 8.13	5.96 ± 13.47	0.29
Patients with endotracheal intubation from IPC, no. (%)	12 (6.5)	12 (6.6)	0.96
Endotracheal intubation from IPC, median ± SD (days)	3 ± 8.16	4.5 ± 6.34	0.62
Patients with mechanical ventilation from IPC, no. (%)	15 (8.1)	15 (8.3)	0.96
Mechanical ventilation from IPC, median ± SD (days)	27 ± 91.37	4 ± 40.25	0.38
Inpatient mortality, no. (%)	31 (17.2)	21 (11.5)	0.15
Infection-related	14 (7.8)	8 (4.3)	0.21
30-day mortality, no. (%)	0 (0)	3 (1.6)	0.08
Infection-related	0 (0)	2 (1.1)	0.15

CHU: conventional hospital unit, ICU: intensive care unit, IPC: index-positive culture.

## Data Availability

All data are available and stored in the informatic tool of our hospital, Clínica Universidad de Navarra.

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
