# Peer review of "Clinical Impact of Rapid Bacterial Microbiological Identification with the MALDI-TOF MS"

_antibiotics, 2023, doi:10.3390/antibiotics12121660_

Round 1
Reviewer 1 Report
Comments and Suggestions for Authors
I have reviewed the manuscript entitled “Clinical impact of rapid bacterial microbiological identification with the MALDI-TOF MS” submitted for possible publication in “Antibiotics-MDPI”. The authors have done great efforts in compiling their findings and have provided a good background as well. Overall, the study is well-conducted and could be beneficial for the reader community. The scientific soundness of the manuscript is fine. However, some of the comments have to be addressed before proceeding it further for possible publication. The manuscript also needs some corrections in the format layout. My specific comments are:
1. The authors are suggested to add more background about the AMR., maybe in one paragraph.
2. In material and methods, the authors are suggested to add the study duration/year.
3. The authors are suggested to add the information about study area (mentioned at line 288-291) at the starting paragraph.
4. If I am the reader, I would love to know about the status of patients, like whether they were admitted in wards or ICUs.
5. The authors are suggested to revise the conclusion part and add some suggestive remarks.
Comments on the Quality of English LanguageThere are some minor checks required.
Reviewer 2 Report
Comments and Suggestions for Authors
The manuscript submitted refers to the use of MALDI-TOF MS for rapid microbiological diagnosis. The authors have designed the study well, however, some considerations need to be taken into account regarding the manuscript:
1) The introduction falls far short of being appropriate for a scientific article. It needs to better describe the problem of the study and the existing gaps, which the authors set out to study.
2) In the methodology, I couldn't find any information that the study had been approved by an ethics committee.
3) The results presented did not make the originality of the study very clear, considering that it is already widely known that MALDI-TOF MS has greatly optimized microbiological diagnosis in clinical routine. Furthermore, the clinical outcomes of the patients were non-significant between the groups studied.
Considering these points and the lack of originality, I believe that the manuscript is not suitable for publication. After making substantial changes and showing what is in fact original in the study, I suggest resubmitting the manuscript.
Reviewer 3 Report
Comments and Suggestions for Authors
Overall, the paper was well-written and explained, however, the lack of novelty is an issue that could be improved. Especially, the CG and IG didn't show much significant difference from the result, and cannot strongly support the final conclusion.
In the title "Clinical impact of rapid bacterial microbiological identification with the MALDI-TOF MS", the author seems to emphasize the method of MALDI-TOF MS, however, the paper, in general, focuses on "early microbiological identification and communication". Especially, in line 167-168, "current study compared the identification information with Vitek 2 in CG and MALDI-TOF MS in IG and reduced 8.7 hours" The author should mention this in the method section in detail, and should have some more discussion about the two methods, how and why MALDI method can reduce time.
Comments on the Quality of English LanguageSome editing issues
Line 27, Line 28, Line 276
Reviewer 4 Report
Comments and Suggestions for Authors
The manuscript describes the clinical impact of early microbiological identification information (in the form of MALDI-TOF MS) on optimizing antibiotic prescribing practices by physicians. This is an extension of previous studies by the same group with the main difference being a shortened time interval (several hours) when microbiological identification information is provided to physicians. It is concluded that promptly providing this information enabled physicians to adjust antibiotic treatment of patients and administer effective oral or combined oral and intravenous therapy. The study can help guide best practices for selecting an appropriate course of action in a clinical setting. A low score for originality/novelty is due to the derivative nature of the work. I have no major suggestions for improvement but these:
Line 253 262 fix abbreviation. You use GC instead of CG for control group
There is no mention of how MALDI samples were obtained or processed. Since this is the main method of analysis there should be some mention of this in the material and methods section.
Round 2
Reviewer 2 Report
Comments and Suggestions for Authors
The authors have made all of the changes cited in the first evaluation of the paper, as well as the changes indicated by other reviewers, in this new version of the manuscript.
The manuscript improved significantly over the initial draft. As a result, I believe it is suitable for publication.